# SELF-CONTROLLER: CONTROLLING LLMS WITH MULTI-ROUND STEP-BY-STEP SELF-AWARENESS

## ABSTRACT

The applications of large language models (LLMs) have been widely spread across all domains. However, the basic abilities such as the controllability of LLMs are still limited. To address this, we propose "**Self-controller**", a novel agentic framework bringing self-awareness into LLMs' reasoning logic. The core idea of this work is to maintain states based on the LLM's response, letting the LLM become self-aware of current status and think step by step in a multi-round chain-of-thought paradigm. Our experiment on the state of textual length has shown the controllability and effectiveness of the Self-controller. We further implement a binary search algorithm to accelerate the generation process based on the linearity and monotonicity of the textual length state. Another advantage of the Self-controller comes with DeepSeek's Context Caching technology, which significantly saves computational token consumption when a cluster of conversations shares the same prefix of context. Theoretically, we prove that in this scenario the extra time complexity is $O(c \log n)$. Results of the back-of-the-envelope estimation suggest that the token consumption of our method is no more than twice as much as that of the trivial single-round generation. Furthermore, our ablation study on word constraints demonstrates the Self-controller's consistent controllability across all foundation models.

## 1 INTRODUCTION

Humans are constantly troubled with self-control issues. Naturally, we ponder the question: do LLMs learning from humans inherit the same characteristics? The willpower of self-control was first discussed in psychology. The study called "the marshmallow test" (Mischel, 2015), was originally developed in the 1960s. The test's goal is to observe how children make decisions between waiting 15 minutes longer for two marshmallows they eagerly wanted or settling for just one right away. The experimental results imply that without accurate reward assessment, the actions of human beings might converge to a local optimum.

The self-awareness mechanism is embedded in the mind. In real practice, to reach a specified goal such as finishing an article with a fixed page size, humans often verify their current progress repetitively. As a result, self-awareness promotes self-control. Adam and Eve "realizes they are naked" in the garden and feels ashamed, starting to wear clothes ever after (Bratcher, 1979). In the meantime, the advancement of LLMs shows new emergent abilities, demonstrating great potential for mimicking such primary intelligent behaviors. However, building self-awareness upon LLMs to ensure controllability is still underexplored.

Length control is a classical task that reflects controllability in LLMs. Traditional length control approaches highly rely on trivial instructions, post-hoc verification, and supervised fine-tuning (SFT). These approaches either lack controllability or are heavily resource-consuming. To address this, we propose a novel framework called "Self-controller". The framework comprises a state reflector and a multi-round dialogue session with LLMs. The reflector engages with LLMs in each round and provides state information on precise statistics of the textual length, as shown in Figure 1. The empirical results on multiple datasets show that our framework

can significantly enhance controllability in LLMs, without significant performance degradation. Based on the linearity and monotonicity of the textual length state, we propose a binary search optimization to achieve efficiency. Combined with DeepSeek's Context Caching technology, we prove that the theoretical extra time complexity is $O(c \log n)$.

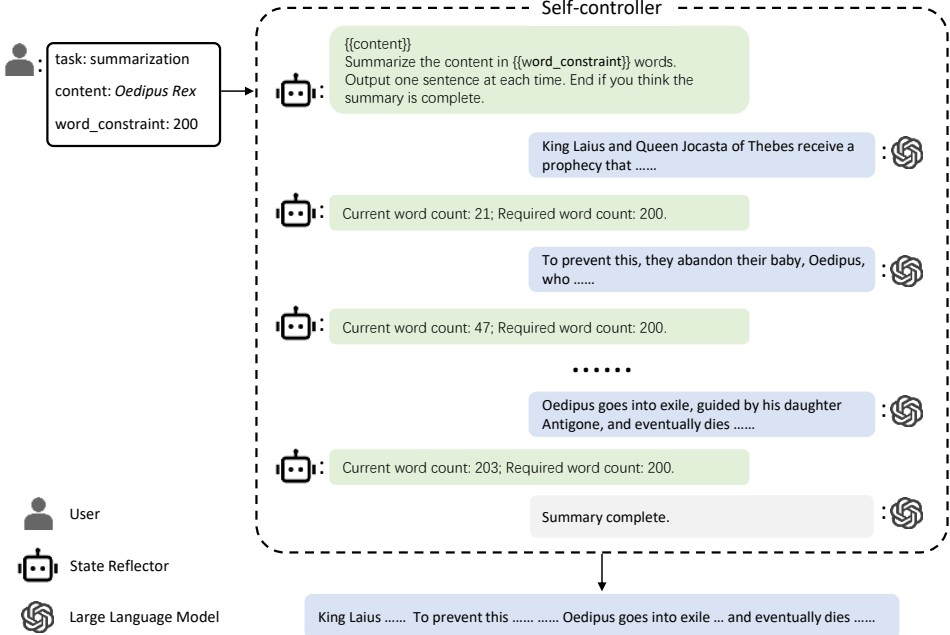

Figure 1: A simplified demonstration of the Self-controller summarizing for "Oedipus Rex"

To sum up, this paper's main contributions are as follows: (1) A novel agentic method "Self-controller" is proposed for length control, containing a multi-round paradigm; (2) We propose the conception of states realizing self-awareness, which applies to any variables as long as it follows reductionism; (3) A binary search algorithm is proposed to speed up the multi-round generation process, reaching an $O(c \log n)$ extra complexity.

## 2 RELATED WORKS

**Large Language Models** In recent years, LLMs' abilities continually grow. Raffel et al. (2020) revealed the broad application potential of large-scale pre-trained models in natural language processing (NLP) tasks through fine-tuning techniques. Brown (2020) demonstrated that even with limited sample sizes, LLMs can still exhibit excellent performance through autoregressive mechanisms. Chowdhery et al. (2023) further explored the relationship between model size and performance, confirming the advantage of larger models in capturing linguistic complexity. With the advent of models such as ChatGPT and GPT-4 (Liu et al., 2023; Achiam et al., 2023), LLMs have drawn even more widespread attention within the field, which can generate text that approaches or even rivals human quality.

**Self-awareness** There are attempts to assess self-awareness in LLMs. Li et al. (2024) propose a unified taxonomy for awareness in LLMs, including capability, mission, emotion, culture, and perspective. Wang et al. (2024) introduces the knowledge quadrant for multimodal LLMs, knowing knowns and unknowns. To enhance trustworthiness and self-awareness in LLMs, various approaches have been proposed, such as Chain-of-thought (Wei et al., 2022; Kojima et al., 2022), Self-consistency (Wang et al., 2023), ReAct (Yao et al., 2023), Tree-of-thought (ToT) (Muralidharan & Thomas, 2024) and Think-Solve-Verify (Liu et al., 2024). LLMs' hallucination problems have been alleviated with these fine-grained stepwise generation frameworks, while the accurate output manipulation remains underexplored.

**Controllability** The trivial method to improve controllability is to include extra zero-shot instructions (such as "*Output in a sonnet style.*") to the prompt (Kojima et al., 2022). Post-hoc verification utilizes additional check programs evaluating LLMs' responses. Chain-of-Verification (CoVe) designs a planning module to raise a set of verification questions and self-corrects them to get the final response afterward (Dhuliawala et al., 2023). Zhang & Gao (2023) applies a similar idea for news claim verification along with the help of the web search. Recently, generating longer text has drawn attention. Bai et al. (2024) proposes AgentWrite, leveraging "divide-and-conquer" agents to generate 10,000+ words, while the length control for each paragraph remains trivial. On the other hand, supervised fine-tuning has been the most prevailing way of ensuring controllability. Juseon-Do et al. (2024) realizes sentence compression with a prompt and improves its performance through instruction-based fine-tuning. Yuan et al. (2024) adds length constraints to the prompt and designs a contrastive proxy task to fine-tune via Reinforcement Learning from Human Feedback (RLHF). Jie et al. (2024) uses reinforcement learning and beam sampling for length control, on GPT-2 models with less than 0.8 billion parameters. Anonymous (2024) embeds length constraints into positional encodings to execute fine-tuning achieving precise length control. The major problems with SFT methods are that they are resource-consuming and lack generalizability.

## 3 METHODOLOGY

### 3.1 OVERVIEW OF THE SELF-CONTROLLER

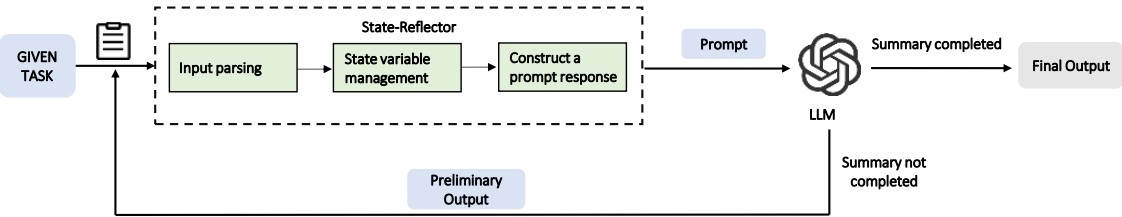

Figure 2: Demonstration of the Self-controller workflow

The abstract workflow of the Self-controller, shown in Figure 2, contains a state reflector submodule. Given a specified task, the Self-controller will maintain relevant state variables in the state reflector. In length control, the state variable is the textual length, making transitions from 0 to the requested word constraint $L_{\text{request}}$. The state reflector parses the new LLM response at each round and updates state variables accordingly. The state is then passed as natural words to the LLM generating the next-step response. The reflection procedure will be recycled indefinitely until the framework gets satisfactory results. In practice, JSON output is necessary for ensuring the stability of the Self-controller, by excluding irrelevant information in the output, especially for weaker foundation models.

## 3.2 Binary Search on the Textual Length State

Based on the linearity and monotonicity of the textual length state, we design the binary-search algorithm for the length control, demonstrated in Algorithm 1. The major difference between the trivial and the binary-search state reflector is the initiative of the state reflector. The trivial state reflector only provides textual length states for the LLM to be self-awareness, including the requesting textual length $L_{\text{request}}$ and the current number of words $\text{len}(S_{\text{output}})$. In binary-search state reflector, the reflector not only provides basic statistics but also actively asks the LLM to output $\frac{1}{2}[L_{\text{request}} - \text{len}(S_{\text{output}})]$ words in the next round, thus executing a binary search on the linear space of $[0, L_{\text{request}}]$. In practice, we set the algorithm to go back to the trivial mode when the surplus textual length is no longer valid for constructing a complete sentence.

---

**Algorithm 1** Binary Search for Length Control

---

**Require:** Initial input $I$, textual length constraint $L_{\text{request}}$, deviation constraint $\delta$.
**Require:** A LLM $LLM$ taking a message list $M$ as input.
**Require:** A state reflector $R$ taking a length parameter $L$ to guide the LLM to output.
**Require:** An empty string to store output $S_{\text{output}}$.
1: $M = \{I\}$
2: **while** $\text{len}(S_{\text{output}}) \leq L_{\text{request}} + \delta$ **do**
3:    $r_1 \leftarrow R\left(\dfrac{L_{\text{request}} - \text{len}(S_{\text{output}})}{2}\right)$
4:    $M \leftarrow M + \{r_1\}$
5:    $r_2 \leftarrow LLM(M)$
6:    **if** "Summary complete" $\in r_2$ **then**
7:       **return** $S_{\text{output}}$
8:    **end if**
9:    $M \leftarrow M + r_2$
10:   $S_{\text{output}} \leftarrow S_{\text{output}} + r_2$
11: **end while**
12: **return** $S_{\text{output}}$

---

## 3.3 Context Caching in Multi-round Sessions

Compared with trivial single-round prompts, the cost of the Self-controller is proportional to the total rounds in the muli-round dialogue. An optimization to this is to build contextual caches, thanks to DeepSeek's **Context Caching** technique, which is made possible by the MLA architecture in DeepSeek-V2 (DeepSeek-AI et al., 2024). Context caching builds textual blocks on disks and saves token consumption by an order of magnitude. When duplicate inputs (only prefixes) are detected, the repeated parts will be retrieved from disks, thus bypassing redundant recomputation.

The theoretical token consumption of 3 different methods analysis is shown in Figure 3. The single-round generation is the trivial prompting method. The multi-round generation is the plain implementation of the Self-controller. The binary search multi-round generation is an optimized version with both Context Caching and binary search on top of the plain Self-controller.

Assume the input length is $L_{\text{input}}$, and the requested textual length constraint is $L_{\text{request}}$. The cost of output tokens is $k$ times as much as input tokens. For simplicity, we ignore the difference in token consumption for each method due to prompting variance, and the difference between the textual length and the number of tokens. For the trivial single-round generation, the overall cost is:

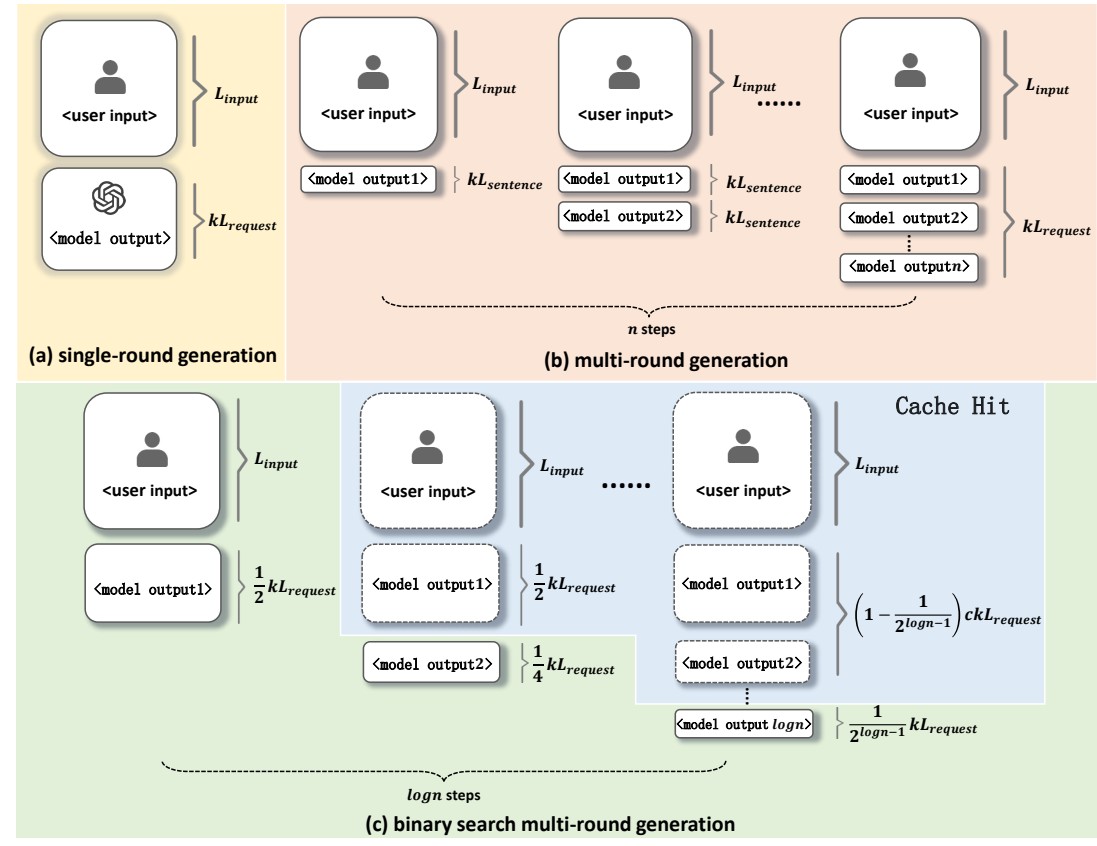

Figure 3: The consumption of 3 different methods: single-round, multi-round, and binary search

$$\text{cost}_{\text{single-round}} = L_{\text{input}} + k \cdot L_{\text{request}} \tag{1}$$

For the multi-round generation, we generate one sentence at a time. The generated textual length in each round is $L_{\text{sentence}}$ on average. We denote the number of total rounds as $n = \dfrac{L_{\text{request}}}{L_{\text{sentence}}}$. The overall cost is:

$$\text{cost}_{\text{multi-round}} = n \cdot L_{\text{input}} + k \cdot \frac{1}{2} \cdot n \cdot (L_{\text{sentence}} + L_{\text{request}}) \tag{2}$$

$$= n \cdot \left[ L_{\text{input}} + \frac{k}{2} \cdot (L_{\text{sentence}} + L_{\text{request}}) \right] \tag{3}$$

For the binary search multi-round generation, the expected generated textual length is proportional to the rest of $L_{\text{request}}$ at each round. $c$ is the cost scaling coefficient due to context caching, meaning the cost of cache-hitting tokens is $c$ times as much as cache-free input tokens ($c < 1$). The number of total rounds is $\log n = \log \dfrac{L_{\text{request}}}{L_{\text{sentence}}}$. The overall cost is:

$$\text{cost}_{\text{binary-search}} = [1 + c(\log n - 1)] \cdot L_{\text{input}} + k \cdot \left[ L_{\text{request}} + c \cdot L_{\text{request}} \cdot \sum_{i=1}^{\log n - 1} \left(1 - \frac{1}{2^i}\right) \right] \tag{4}$$

$$= [1 + c(\log n - 1)] \cdot L_{\text{input}} + k \cdot L_{\text{request}} \cdot \left[ \left(1 + c \left( \log n - 1 - \sum_{i=1}^{\log n - 1} \frac{1}{2^i} \right) \right) \right] \tag{5}$$

$$= [1 + c(\log n - 1)] \cdot (L_{\text{input}} + k \cdot L_{\text{request}}) - k \cdot L_{\text{request}} \cdot c \cdot \sum_{i=1}^{\log n - 1} \frac{1}{2^i} \tag{6}$$

$$= [1 + c(\log n - 1)] \cdot \text{cost}_{\text{single-round}} - k \cdot L_{\text{request}} \cdot c \cdot \sum_{i=1}^{\log n - 1} \frac{1}{2^i} \tag{7}$$

$$< (1 + c \log n) \cdot \text{cost}_{\text{single-round}} \tag{8}$$

The result shows that the extra time complexity is $O(c \log n)$ compared with the trivial method. Let's have a back-of-the-envelope estimation. In practice, long output length such as Bai et al. (2024) can generate $L_{\text{request}} \approx 10,000$ words. On average, the textual length of natural sentences has $L_{\text{sentence}} < 20$. For DeepSeek, $c = 0.1$. Therefore, we have

$$c \log n = c \log \frac{L_{\text{request}}}{L_{\text{sentence}}} \approx 0.1 \times \log \frac{10,000}{20} \approx 0.1 \times 8.96 < 1 \tag{9}$$

which indicates

$$\text{cost}_{\text{binary-search}} < (1 + c \log n) \cdot \text{cost}_{\text{single-round}} < 2 \cdot \text{cost}_{\text{single-round}} \tag{10}$$

This proves the efficiency of the binary search multi-round approach.

## 4 EXPERIMENTS

### 4.1 LENGTH CONTROL

In this experiment, 3 datasets are selected from HuggingFace: *webis/tldr-17* (short as "tldr"), *argilla/news-summary* (short as "news"), and *ccdv/arxiv-summarization* (short as "arxiv"). For each dataset, we sample 128 instances with textual lengths ranging in $[800, 1200]$ words. For foundation models, we choose the following models: GLM series (GLM-4-air, GLM-4-flash), GPT series(GPT-4o, GPT-4o-mini, GPT-3.5-turbo), and Deepseek-V2. The word constraint $L_{\text{request}}$ is 250 words. Results are within Table 1.

To ensure the textual generation quality, we evaluate the former results on BERTSCORE proposed by Zhang et al. (2020). We introduce a hypothesis here that the ground-truth reference is the generated text by GPT-4o in single-round respectively. Therefore, the BERTSCOREs of GPT-4o in single-round equals 1 on all datasets. The evaluation results are illustrated in Figure 4, which shows no significant textual quality loss on most models. The degradation and low performance in Table 1 on GLM-4-flash may indicate the inner inability of weaker models. The gap between GPT-4o and all other models may be caused by fine-grained writing styles.

| Dataset | tldr | | | news | | | arxiv | | |
|---|---|---|---|---|---|---|---|---|---|
| **Model** | $\lvert\Delta\rvert$ | STD | ratio | $\lvert\Delta\rvert$ | STD | ratio | $\lvert\Delta\rvert$ | STD | ratio |
| GLM-4-flash single-round | 122.43 | 22.15 | 17.4% | 104.30 | 34.20 | 23.5% | 92.45 | 28.70 | 18.2% |
| GLM-4-flash multi-round | 58.41 | 95.05 | 49.6% | 62.44 | 94.26 | 50.3% | 46.27 | 78.95 | 38.8% |
| GLM-4-air single-round | 86.61 | 31.62 | 19.4% | 73.77 | 43.83 | 24.9% | 54.06 | 31.31 | 16.0% |
| GLM-4-air multi-round | 80.58 | 39.18 | 23.1% | 59.16 | 62.79 | 32.9% | 69.33 | 34.22 | 18.9% |
| GPT-3.5 single-round | 127.09 | 18.23 | 14.8% | 100.09 | 17.69 | 11.8% | 106.73 | 18.35 | 12.8% |
| GPT-3.5 multi-round | 37.21 | 45.82 | 21.5% | 1.63 | 17.61 | 7.0% | 8.91 | 33.56 | 13.9% |
| GPT-4o-mini single-round | 38.70 | 29.64 | 14.0% | 23.21 | 11.11 | 4.9% | 26.27 | 11.45 | 5.1% |
| GPT-4o-mini multi-round | 6.27 | 23.68 | 9.2% | 10.13 | 7.67 | 2.9% | 8.32 | 20.92 | 8.1% |
| GPT-4o single-round | 16.32 | 12.56 | 5.4% | 0.19 | 23.79 | 9.5% | 2.52 | 12.71 | 5.1% |
| GPT-4o multi-round | 6.42 | 6.17 | 2.4% | 6.85 | 23.56 | 9.2% | 5.39 | 16.26 | 6.4% |
| DeepSeek-chat single-round | 24.38 | 31.30 | 13.9% | 10.61 | 35.72 | 13.7% | 11.28 | 32.29 | 12.4% |
| DeepSeek-chat multi-round | 2.66 | 8.52 | 3.4% | 4.95 | 24.09 | 9.4% | 6.41 | 20.58 | 8.0% |

Table 1: Length control results across different datasets and LLMs (word constraint = **250**). Grey and green color represent better performance between single-round and multi-round separately for each model. Box indicates the best result in each column.$\lvert\Delta\rvert$ is the absolute value of the difference between the average output textual length (AVG) and $L_{\text{request}}$. STD is the standard deviation of output textual lengths. And ratio = STD/AVG

.

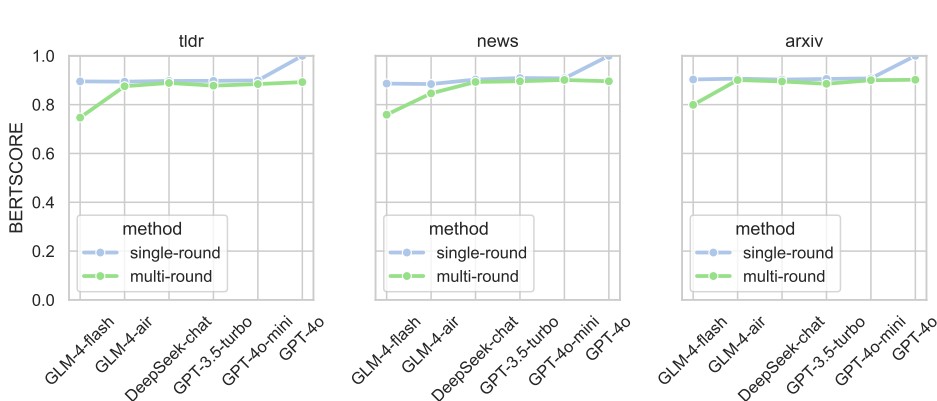

Figure 4: Average BERTSCORE evaluation across different datasets and LLMs (word constraint = **250**)

## 4.2 BINARY SEARCH ON LENGTH CONTROL

To study the effectiveness of the binary-search method, we select 128 samples with longer textual lengths from the *tldr* and the *arxiv* dataset, ranging in $[2000, 2500]$ words. The *news* dataset doesn't contain enough satisfiable samples thus being excluded in this section. The word constraint is set to 500 words. The results

are shown in Table 2. For most cases, the binary-search method achieves better results evidentially, proving our optimization's effectiveness. One thing worth mentioning is that DeepSeek-chat maintains strong consistency on all settings both in Table 1 and Table 2.

| Dataset | | tldr | | | arxiv | | |
|---|---|---|---|---|---|---|---|
| **Model** | | $|\Delta|$ | STD | ratio | $|\Delta|$ | STD | ratio |
| GLM-4-flash | single-round | 171.71 | 73.24 | 22.3% | 112.13 | 43.57 | 11.2% |
| | binary-search | 22.17 | 92.72 | 19.4% | 31.92 | 99.80 | 21.3% |
| GLM-4-air | single-round | 152.63 | 66.99 | 19.3% | 90.55 | 55.50 | 13.6% |
| | binary-search | 30.75 | 100.52 | 21.4% | 27.43 | 57.09 | 12.1% |
| GPT-3.5-turbo | single-round | 330.47 | 60.41 | 35.6% | 203.71 | 49.46 | 16.7% |
| | binary-search | 43.46 | 14.82 | 3.2% | 38.88 | 22.19 | 4.8% |
| GPT-4o-mini | single-round | 60.46 | 48.25 | 11.0% | 13.79 | 38.72 | 7.5% |
| | binary-search | 16.20 | 7.82 | 1.5% | 12.29 | 114.96 | 23.6% |
| GPT-4o | single-round | 52.17 | 49.41 | 11.0% | 22.92 | 40.85 | 8.6% |
| | binary-search | 11.47 | 6.68 | 1.3% | 3.31 | 61.47 | 12.2% |
| DeepSeek-chat | single-round | 155.24 | 137.80 | 40.0% | 38.69 | 82.88 | 15.4% |
| | binary-search | 28.13 | 40.41 | 8.6% | 11.01 | 28.47 | 5.8% |

Table 2: Length control results across different datasets and LLMs (word constraint = **500**)

### 4.3 ABLATION STUDY

**Efficiency**   We examine the efficiency between the multi-round generation and binary search multi-round generation. We choose 64 samples from the *tldr* dataset ranging in $[2000, 2500]$ words, along with the representative word constraints as $[200, 400, 600, 800, 1000]$. Results are illustrated in Figure 5. For weaker LLMs (GPT-3.5-turbo and GLM-4-flash), the efficiency is insignificant in terms of dialogue rounds. For stronger LLMs with moderate instruction following ability, the number of rounds for multi-round generation grows rapidly with the growth of word constraints, while the binary-search version grows slowly at a near log scale. This empirical finding aligns with the theoretical results.

**Word Constraint**   As observed in former experiments, the overall performance of the Self-controller varies according to different word constraint parameters. In this part, we carefully investigate the influence of word constraints. we set the word constraint within a dynamic range of $[50, 1300]$. All other settings are equivalent to the ablation study on efficiency. The results are shown on scatter plots in Figure 6. This clearly shows that GPT-series are gaining immanent controllability with OpenAI's pretraining advancement on foundation models, which counteracts the Self-controller's controlling effect on small constraints ($L_{\text{request}} < 500$). However, the results imply that the Self-controller takes effect on all word constraints and all models, demonstrating its substantial generalizability.

## 5   DISCUSSION

**Exploring Automatic State Management**   Although the states are not confined to any specific task, the self-controller's prompts still require revision for each new scenario to manage novel states effectively. For future work, we can develop real-time state management based on user intentions, utilizing a society of

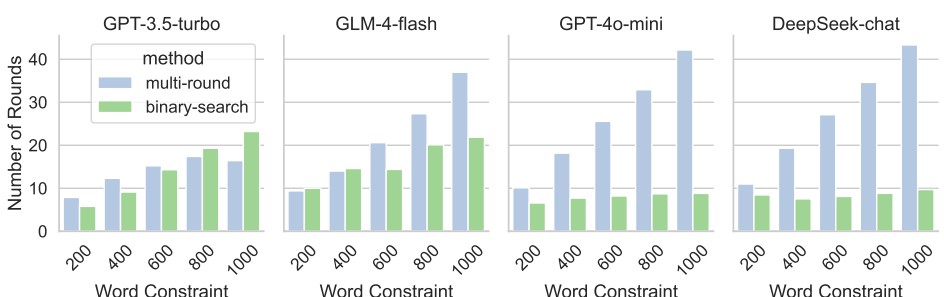

Figure 5: Efficiency study for the binary search multi-round generation method on the tldr dataset

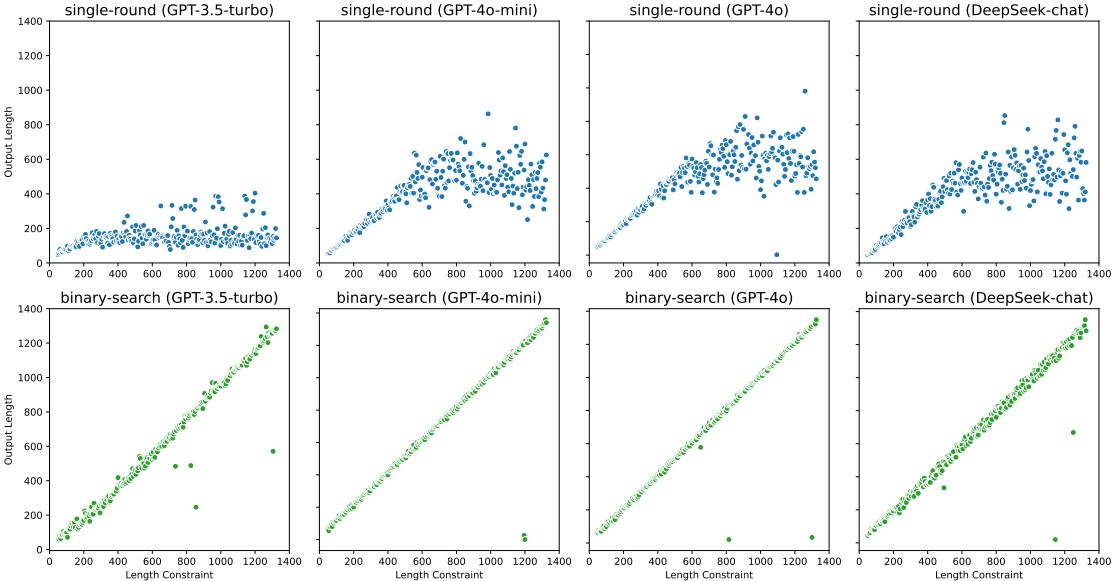

Figure 6: Ablation study on word constraint

agents. Inspired by the binary-search optimization, more heuristic reflections may accelerate the convergence of state transitions.

**Enhancing Prompt Engineering Techniques** Improving output quality is a key area for further development. An additional refinement process for the self-controller's final output can enhance textual quality and detail expression. For shorter constraints $L_{\text{request}} < 500$, advanced foundation models like GPT-4o already perform well. However, longer outputs may benefit from planning techniques such as the divide-and-conquer approach proposed by Bai et al. (2024). In the future, exploring the integration of planning within the multi-round paradigm of the self-controller could provide global controllability on a book-scale.

**New Paradigm of Thoughts** Towards the dawn of reasoning, ReAct and ToT represent two paths for future directions on CoT. The major difference between these two paths is the way in which the message is passed. ReAct can reason within a single session, while ToT requires assembly in different sessions. Recently, Diagram-of-thought (DoT) (Zhang et al., 2024) proposes a similar idea to our Self-controller, executing reasoning in multi-round sessions. With the development of near-infinite input and output length of LLMs, the multi-round sessions can be more informative than a cluster of generated responses in the future.

## 6 CONCLUSION

In this paper, we introduced the Self-controller, a novel framework designed to enhance the controllability of LLMs by incorporating self-awareness into their reasoning processes. By maintaining state variables and enabling multi-round dialogue sessions, the Self-controller allows LLMs to refine their outputs iteratively, achieving more precise control over tasks such as length control. Our experiments demonstrated that the Self-controller significantly improves the controllability of LLMs across various datasets and models, without compromising the quality of the generated text. Implementing binary search optimization further enhances the efficiency of the multi-round generation process, reducing the computational overhead. The results also suggest that the Self-controller is adaptable to different foundation models and various word constraints, showcasing its generalizability and robustness. Future work will focus on automating state management and exploring more advanced prompt engineering techniques to enhance output quality. Integrating planning techniques within the multi-round paradigm could also provide global controllability for more complex tasks. In conclusion, the Self-controller represents a significant step towards building more controllable and self-aware LLMs, paving the way for more reliable and versatile applications in natural language processing.

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
