# OpenReview forum: "Self-controller: Controlling LLMs with Multi-round Step-by-step Self-awareness"
_ICLR.cc/2025/Conference — ICLR 2025 Conference Withdrawn Submission_

### Official Review · Reviewer_myqf · 2024-10-17

**Soundness:** 2
**Presentation:** 2
**Contribution:** 2
**Rating:** 3
**Confidence:** 3

**Summary:**

This paper proposes 'self-controller,' a new framework that brings self-awareness into LLMs' reasoning logic. Specifically, the authors use multiple rounds of conversations to make the output length of LLMs more controllable. In addition, a binary search algorithm is implemented along with DeepSeek's context caching technology to reduce the complexity of the proposed method. Experimental results show that the proposed methods can better control the output length without significant textual quality loss.

**Strengths:**

This method can effectively control the output length of LLMs.

**Weaknesses:**

1. I feel this paper is over-claiming. Controllability of LLMs is a very broad research topic, but the study in this paper mainly focuses on controlling the output length of LLMs, which is a very small part of this research area. This narrows down the applications of the proposed methods to other areas, such as sentiment control, keyword control, etc.

2. There are basically no baselines except for the trial single-round baseline. Again, more controllability aspects of LLMs should be studied to demonstrate the effectiveness of the proposed methods.

**Questions:**

N/A

---

### Official Review · Reviewer_jRiq · 2024-10-20

**Soundness:** 3
**Presentation:** 2
**Contribution:** 2
**Rating:** 3
**Confidence:** 3

**Summary:**

The paper presents Self-controller, a novel framework for improving the controllability of large language models by incorporating self-awareness into their reasoning processes. The framework maintains state variables and iteratively refines outputs through multi-round dialogue sessions, allowing LLMs to adjust their responses dynamically. It also leverages a binary search algorithm for textual length control, enhancing efficiency in generating content of a specific length. Furthermore, the DeepSeek’s Context Caching technology is integrated to optimize token consumption, reducing computational overhead. The framework demonstrates effectiveness in controlling text length across various datasets and models without significant degradation in text quality.

**Strengths:**

1. The Self-controller introduces a novel idea by applying self-awareness mechanisms to control LLM behavior in a multi-round step-by-step manner.
2. The paper offers a detailed explanation of the methodology and providing empirical evidence of the system’s effectiveness across different models and datasets.
3. The paper is generally well-written and structured, with clear diagrams and explanations.

**Weaknesses:**

1. This paper lacks of the comparison with other baselines. The authors need to add more baselines (as described in related work) to compare, this could better show the effectiveness of the proposed method.
2. This paper lacks of the qualitative examples demonstrations, which could make the reader better understand the task and method. I suggest the authors put some examples to the appendix at least.

**Questions:**

The double quotation marks looks strange in the paper, the author may change the double quotation marks.

---

### Official Review · Reviewer_aLnP · 2024-11-03

**Soundness:** 3
**Presentation:** 3
**Contribution:** 2
**Rating:** 5
**Confidence:** 3

**Summary:**

In this paper, the authors propose a novel framework called Self-controller for length control in LLMs. The main idea in the paper is is to maintain states based on the LLM’s response, letting the LLM become self-aware of current status and think step by step in a multi-round chain-of-thought paradigm. More specifically, the framework comprises a state reflector and a multi-round dialogue session with LLMs. The reflector engages with LLMs in each round and provides state information on precise statistics of the textual length. Based on the linearity and monotonicity of the textual length state, the authors propose a binary search optimization to achieve efficiency. The authors also obtain theoretical guarantees for the extra time complexity.

**Strengths:**

Strengths:

1.	The use of a binary search algorithm combined with the state reflector for length control shows an effective approach for optimizing response generation in LLMs.

2.	Context caching method reduces the cost for multi-round sessions, increasing the real-world applicability.

**Weaknesses:**

Weaknesses:

1.	It seems like the weaker models such as GLM4-flash and GLM-4-air do not benefit much from the proposed multi-round strategy.

2.	The methodology is novel, however, I believe the contribution is not enough for an ICLR acceptance. Please see the question below.

**Questions:**

Questions:

1.	Can authors elaborate on the novelty of their method as compared to the existing length control baselines? Is the state reflector module the main novelty?

2.	Can authors elaborate a little bit more on the hypothesis mentioned in row 276? What is the purpose of this hypothesis?

3.	Why is it that the weaker models such as GLM-4-flash and GLM-4-air do worse in multi-round than in the single round? (See Figure 4)

---

### Official Review · Reviewer_gf23 · 2024-11-04

**Soundness:** 2
**Presentation:** 2
**Contribution:** 2
**Rating:** 3
**Confidence:** 4

**Summary:**

The authors present Self-controller, a multi-round prompting approach to more efficiently guide LLMs to generate text with length constraints. The authors claim their framework can improve the self-awareness and self-control of LLMs.

The authors apply Self-controller on the task of length-constrained text generation, but not to any other tasks, and so it is not known whether their approach would generalize to other tasks and actually improve the self-reflection of LLMs more generally. How does high performance on this task imply self-awareness? Wouldn't multi-round prompting without binary search alone be sufficient to demonstrate "self-awareness"?

The prompting approach is described as a binary search, but this analogy isn't very clear: What is the item that the search algorithm is aiming to find?

**Strengths:**

- The authors present a new prompting approach that guides LLMs to more accurately meet the length constraint in a length-constrained generation task.
- The proposed method is relatively simple and could theoretically be extended to other tasks that would benefit from the model's awareness of task state information.

**Weaknesses:**

- The claim that the proposed approach facilitates self-reflection is overly broad. The authors only demonstrate an improvement on the task of length-constrained generation, and it is not known to what extent their results generalize to other tasks that benefit from self-reflection.
- The cost model is assumed to be that of an LLM service via an API, where the cost of each prompt is given by the number of input tokens plus output tokens (possibly with each input token being more/less expensive than an output token). But this is not explicitly stated, and there is no comparison to the cost in a local setting, where the model's cost is more determined by the number of forward passes.
- The prompting approach is described as a binary search, but this analogy is unclear/imprecise.

**Questions:**

More detailed feedback is listed below. I also include grammatical and stylistic errors but only for the first two sections, and this list is not comprehensive, so I strongly encourage the authors to carefully re-read the manuscript and correct all similar errors. Research questions/comments about the content for all sections are included below.

Line 29: "Humans are constantly troubled with self-control issues."
  How so? Example? Citation?

General: (minor) Use `` for left double quotation marks rather than ".

Line 33: "The experimental results imply that
without accurate reward assessment, the actions of human beings might converge to a local optimum"
  How so? Do all children end up picking one option if the delay is not set appropriately?

Line 37: Adam and Eve, being non-historical figures, don't really serve as good supporting evidence for the claim that self-awareness is important in human intelligence.

Line 37: "realizes" -> "realize" (grammatical number agreement)

Line 40: "building self-awareness upon LLMs"
  It isn't totally clear what the precise definition of self-awareness and self-control is.

Line 49: "DeepSeek’s Context Caching"
  Needs citation.

Line 50: "extra time complexity is O(c log n)"
  What is c? What is n?

Line 101: "the accurate output manipulation" -> "accurate output manipulation" or "the accurate manipulation of output"

Line 108: "the web search" -> "web search"

Line 145: "initiative of the state reflector"
  What does "initiative" mean here?

Section 3.3 and Figure 3: What does "consumption" mean here? What is the cost measured in, here? Number of prompts? Number of forward passes/FLOPs? Reading further, I think the cost is in terms of the numbed of input and output tokens (as would be charged in the use of an API for an LLM). But would there be any difference in cost if the LLM is running locally (i.e., in terms of number of forward passes)? I think further clarification is required, as well as a comparison with the cost in the non-API setting.

Why only test on text lengths between 800 and 1200? Why not try much shorter and longer lengths? Similarly, why not try shorter lengths in the binary search experiments?

Are there any experiments validating the use of BERTSCORE in this task? For example, human evaluation of a subsample of outputs?

Table 2: Why not compare binary-search with multi-round?

Line 423: "dawn of reasoning"
  This is highly vague and abstract. I assume the authors are referring to the time at which prompting methods were developed that significantly enhanced LLM performance on reasoning-intensive tasks. I strongly suggest re-wording and avoiding the use of "dawn."

Line 427: "near-infinite"
  Though context sizes have increased, they are still quite a ways off from infinite. If what was intended was that the context size is sufficiently large for most inputs, then I suggest rewording this to be more descriptive, and avoiding comparisons to infinity.

---

### Note · Authors · 2024-11-28

I have read and agree with the venue's withdrawal policy on behalf of myself and my co-authors.